



# On the relative roles of dynamics and chemistry governing the abundance and diurnal variation of low latitude thermospheric nitric oxide

David E. Siskind[1], McArthur Jones, Jr.[1], Douglas P. Drob[1], John P. McCormack[1], Mark E. Hervig[2], Daniel R. Marsh[3], Martin G. Mlynczak[5], Scott M. Bailey[4], Astrid Maute[3], and Nicholas J. Mitchell[6]

[1]Space Science Division, Naval Research Laboratory, Washington, DC
[2]GATS Inc., Driggs ID
[3]NCAR, Boulder CO
[4]Virginia Polytechnic, Blacksburg VA
[5]NASA Langely Research Center, Hampton VA
[6]Centre for Space, Atmospheric and Oceanic Science, University of Bath, UK

**Correspondence:** David Siskind (david.siskind@nrl.navy.mil)

**Abstract.** We use data from two NASA satellites, the Thermosphere Ionosphere Energetics and Dynamics (TIMED) and the Aeronomy of Ice in the Mesosphere (AIM) satellites in conjunction with model simulations from the Thermosphere-Ionosphere-Mesosphere-Electrodynamics General Circulation Model (TIME-GCM) to elucidate the key dynamical and chemical factors governing the abundance and diurnal variation of nitric oxide (NO) at near solar minimum conditions and low

latitudes. This analysis was enabled by the recent orbital precession of the AIM satellite which caused the solar occultation pattern measured by the Solar Occultation for Ice Experiment (SOFIE) to migrate down to low and mid latitudes for specific periods of time. We use a month of NO data collected in January 2017 to compare with two versions of the TIME-GCM, one driven solely by climatological tides and analysis-derived planetary waves at the lower boundary and free running at all other altitudes, while the other is constrained by a high-altitude analysis from the Navy Global Environmental Model (NAVGEM)

up to the mesopause. We also compare SOFIE data with a NO climatology from the Nitric Oxide Empirical Model (NOEM). Both SOFIE and NOEM yield peak NO abundances of around $4 \times 10^7$ cm$^{-3}$; however, the SOFIE profile peaks about 6-8 km lower than NOEM. We show that this difference is likely a local time effect; SOFIE being a dawn measurement and NOEM representing late morning/near noon. The constrained version of TIME-GCM exhibits a low altitude dawn peak while the model that is forced solely at the lower boundary and free running above does not. We attribute this difference due to a

phase change in the semi-diurnal tide in the NAVGEM-constrained model causing descent of high NO mixing ratio air near dawn. This phase difference between the two models arises due to differences in the mesospheric zonal mean zonal winds. Regarding the absolute NO abundance, all versions of the TIME-GCM overestimate this. Tuning the model to yield calculated atomic oxygen in agreement with TIMED data helps, but is insufficient. Further, the TIME-GCM underestimates the electron density [e-] as compared with the International Reference Ionosphere empirical model. This suggests a potential conflict with

the requirements of NO modeling and [e-] modeling since one solution typically used to increase model [e-] is to increase the solar soft X ray flux which would, in this case, worsen the NO model/data discrepancy.



# 1 Introduction

Nitric oxide (NO) has long been recognized as one of the most important trace constituents in the middle and upper atmosphere. This is due to its role in cooling the thermosphere through mid-IR emission (Kockarts, 1980; Mlynzcak et al., 2003; Knipp et al., 2017), as a source for $NO^+$ ions in the lower ionosphere (Solomon et al., 2006) and more generally as an indicator of energy

input into the atmosphere (Siskind et al., 1989b; Barth et al., 1999; Mlynczak et al., 2018a). Motivated by the development of whole atmosphere models and the availability of new datasets, there has been much recent work on the properties and role of NO at high latitudes. Here NO can serve both as measure of energetic particle precipitation (EPP) into the atmosphere (Hendrickx et al., 2015; 2018; Smith-Johnson, 2017) and as a tracer for descent in the winter polar vortex (Newnham et al., 2018; Siskind et al., 2015; Randall et al., 2015; Bailey et al., 2014) and ultimately coupling with the chemistry of the

mesospheric and stratosphere (Funke et al., 2017). There has been less recent work on low latitude nitric oxide, although it has recently been shown that equatorial NO can be used as a diagnostic of non-migrating tides (Oberheide and Forbes, 2008; Oberheide et al., 2013). The primary emphasis of this work will be on equatorial NO, its absolute abundance and its diurnal variability.

Most of the extant thermospheric NO measurements are limited to specific local times. This results either from satellites

in sun-synchronous orbits or from satellites which might be in varying local time orbits, but which use a technique such as solar occultation which is inherently limited to a single local time. Examples of the first case include the Student Nitric Oxide Explorer (SNOE, Barth et al., 1998), the Michelson Interfrerometer for Passive Atmospheric Sounding (MIPAS) dataset on the European research satellite Envisat (Bender et al., 2015; Bermejo-Pantaleón et al., 2011) and the sub millimeter radiometer (SMR) on the Swedish Odin satellite (Kiviranta et al., 2018; Sheese et al., 2013). Examples of the second case include the

Solar Occultation for Ice Experiment (SOFIE) (Gomez-Ramirez et al., 2013) on the Aeronomy of Ice in the Mesosphere (AIM, Russell et al., 2009), the ACE Fourier Transform Spectrometer (ACE-FTS) (Bernath et al., 2005; Bender et al., 2015) and the Halogen Occultation Experiment (HALOE) data on the NASA UARS satellite (Siskind et al., 1998; Russell et al., 1993). We thus deduce that unfortunately there is no satellite data which can directly resolve the diurnal variation of thermospheric nitric oxide. Therefore any exploration of this variation must necessarily be indirect. Here, with the assistance of a thermospheric

general circulation model, we present such an indirect approach.

Specifically, we compare SOFIE data with data from SNOE as encapsulated in the Nitric Oxide Empirical Model (NOEM, Marsh et al., 2005). As noted above SOFIE measures at either sunrise or sunset; SNOE data were acquired at about 1100 local time. To compare the two we will use diurnally resolved model results from the Thermospheric Ionospheric Mesosphere Electrodynamics General Circulation Model (TIME-GCM) as recently described by Jones et al. (2018). One additional, important

aspect to our model data comparisons are that they are multi-constituent. To understand how the diurnal variation of nitric oxide might be sensitive to migrating tidal amplitudes, we will compare our model results with observations of upper mesospheric zonal winds (deWit et al., 2013). Further, since the abundance of NO is known to be sensitive to atomic oxygen (Siskind et al., 1989a), we will compare TIME-GCM output to Sounding of the Atmosphere with Broadband Emission Radiometry (SABER) atomic oxygen ($O$) data. Finally, since the ionization that ultimately leads to nitric oxide also produces the E-region ionosphere



(Solomon, 2001; Sojka et al., 2013), we will compare our model output to the International Reference Ionosphere (IRI; Bilitza, 2015). In this manner, we will significantly reduce the number of free parameters to guide future model development.

## 2 Overview of Data

For most of the AIM mission, the SOFIE occultations have been confined to high latitudes, consistent with the focus of AIM on Polar Mesospheric Clouds. However, recently, due to the precession of the AIM orbit, this occultation pattern has occasionally migrated to lower latitudes. Of specific interest for this work is the period from December 2016-January 2017 where SOFIE occultations were confined to near equatorial latitudes. To quantitatively analyze this period we are forced to make some key assumptions about the causes of NO variability in the tropics. This is because the TIME-GCM model we use is constrained by meteorological data from January 2010, not January 2017. However, at tropical latitudes, when averaged over a month, it is reasonable to assume that the average NO will be governed by solar and geophysical forcing. Figure 1 shows the variation of the solar and geomagnetic activity indices for both months. It shows that for both months, solar activity was, on average, low, although slightly higher than absolute solar minimum. The average F107 was 78 for 2010 and 75 for 2017. Likewise, geomagnetic activity was equally low: averaged Ap was 9 for 2017 and 3 for 2010. These differences are very small and thus we argue that while the day-to-day NO might vary due to meteorological forcing from below, when averaged over a month, there should be little difference in the NO profiles for January 2010 and 2017.

Figure 2 shows the monthly averaged SOFIE profile compared with the NOEM results computed for 2010 and 2017. First, the NOEM results for the two years are almost identical and this supports our arguments above about the acceptability of comparing the two years. Second, both SOFIE and NOEM give peak NO densities of about $4.5 \times 10^7$ cm$^{-3}$; however, the peak in the SOFIE NO is displaced downward by about 8 km from NOEM (note: The NOEM profile peaks at 110 km; an examination of individual SNOE profiles, from which NOEM is derived, often show the peak altitude closer to 108 km (not shown). Regardless, this difference between SOFIE and SNOE/NOEM of either 6 or 8 km is significantly greater than the 2-3 km altitude resolution of either instrument). As we will discuss below, this altitude difference likely reflects the local time difference between the SOFIE data (5-6 AM) and SNOE, as encapsulated by NOEM (near 11 AM). Above the peak, SOFIE NO is lower than NOEM, but at the higher altitudes, above 120 km, SOFIE appears to approach the NOEM values. These three facets of the data, the peak magnitude, the altitude of the peak and the behavior at the higher altitudes will be the subject of the model-data comparisons below.

## 3 Model-data comparisons

### 3.1 Overview of approach

We have adapted the TIME-GCM so that we can compare two approaches towards modeling the mesosphere and lower thermosphere (MLT). The standard version of the TIME-GCM uses a combination of the Global Scale Wave Model (GSWM, Zhang et al., 2010a and b) as a bottom boundary (approximately 30 km) for the diurnal and semi-diurnal tidal forcing and the

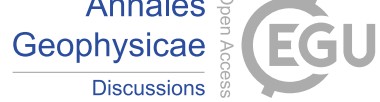



daily averaged European Center for Medium Range Weather Forecast (ECMWF) analysis for the planetary wave distribution. We will herein call this the "driven" model. We will compare this driven model with a version of the model where it is nudged to the winds and temperatures from the high altitude version of NAVGEM (Navy Global Environmental Model, described in McCormack et al. (2017)). The nudging technique is described more fully by Jones et al, (2018). As they discuss, at the

lower boundary of the TIME-GCM, the nominal GSWM and ECMWF fields are replaced with the NAVGEM analyses using a normalized weighting factor of unity. This weighting factor decreases with increasing altitude and becomes zero at the top of the NAVGEM analysis, at approximately 95 km. Above this altitude, the TIME-GCM is free running.

In addition, to more fully explore the possible roles of dynamics and chemistry on the NO, we performed two additional simulations with the NAVGEM-nudged model where we perturbed some key input parameters. One variation was to reduce

the vertical eddy diffusion (Kzz) coefficient by a factor of 10; the other was to increase the quenching of metastable atomic nitrogen ($N^2D$) by a factor of 2. The rationale behind both these changes was to reduce the calculated NO abundance, which as we show below, is too large in the baseline cases. This can be understood by considering the chemistry of $N(^2D)$. In the MLT, $N(^2D)$ can react either with molecular oxygen to produce NO according to

$$N(^2D) + O_2 \rightarrow NO + O \tag{1}$$

or it can be quenched by atomic oxygen according to

$$N(^2D) + O \rightarrow N(^4S) + O \tag{2}$$

Equation (2) is then followed by

$$N(^4S) + NO \rightarrow N_2 + O \tag{3}$$

which is the ultimate sink of nitric oxide. Thus to reduce the calcuated NO, we aim to increase the rate of reaction (2) at the

expense of reaction (1). As we show below, we did this by either increasing the calculated atomic oxygen, or by increasing the reaction rate coefficient governing equation (2). The standard TIME-GCM model uses a value for (2) of $7 \times 10^{-13}$ cm$^{-2}$s$^{-1}$ from Fell et al. [1990]. As discussed by Yonker[2013], Herron [1999] appears to neglect this reference and instead recommends a room temperature value which is about double the Fell et al. [1990]. Thus in doubling the rate coefficient reaction (2) we are essentially using the room temperature Herron value in lieu of the Fell et al. [1990] value.

Figure 3 compares the diurnal NO variation between the driven and nudged models. It shows that both models have peak NO in the lower thermosphere of just over $10^8$ cm$^{-3}$. The nudged model peak occurs at sunrise, while the driven model's peak occurs at midnight. Two points are evident from these simulations. First, it is apparent that the overall magnitude of the peak NO density is much greater than the observations. Second, the peak NO abundance in the nudged model occurs almost precisely at the local time of the SOFIE measurements. After about 0800 local time, into the early afternoon, the altitude of the

peak rises several kilometers. This seems qualitatively consistent with the difference between SOFIE and NOEM. By contrast, for the driven model, the peak NO is at midnight and there is no change in altitude between sunrise and early afternoon. To attempt to see how diurnal variation in the two models might compare with the SOFIE/NOEM difference, we took the ratio of





the sunrise NO in the models (average of 0500-0600 localtime) with the 11 AM profiles (SNOE local time) and plot them with
the ratio of SOFIE/NOEM. This is shown in Figure 4.

The figure shows that the SOFIE/NOEM ratio is greatest at 100 km (about a factor of 1.7) and decreases monotonically up
to about 115 km. Neither model exactly reproduces this behavior, but the slope in nudged model between 115 and 105 km

comes much closer. Like the data, the nudged model decreases with increasing altitude between 105 and 115 km whereas the
driven model ratio is nearly constant in this altitude range. The absolute value of the sunrise/11 AM ratio in the nudged model
is also in better agreement than the driven model; it shows a peak at about 105 km of about 1.4 and decreases above. This is
consistent with the change in peak altitude seen in Figure 3. The ratio in the driven model barely exceeds 1.0, consistent with
no change in the peak altitude between 0500 and 1100.

Also of interest is that neither model shows the sharp turnaround in the sunrise/11 AM ratio seen in the SOFIE/NOEM
comparison above 115-120 km. This turnaround reflects the fact that the SOFIE values are approaching NOEM at these
altitudes, while both models suggest that the dawn values should be much lower than those nearer noon. Since the NO variation
in the model above 120 km is driven solely by nighttime recombination of NO with $N(^4S)$ according to reaction (3) above, it
is difficult to imagine a scenario whereby the dawn NO should be larger than mid-day. This change in slope could be driven

by lower signal to noise in the retrieval at the higher altitudes; however, the SOFIE curve in Figure 2 represents an average of
about 400 profiles which should reduce the noise effects. It could also result from an as-yet-unidentified bias at these altitudes.
More work will be needed to clarify this.

### 3.2 Absolute Magnitude

The analysis above suggests that the difference between the nudged and driven model may shed light on the differences between

SOFIE and SNOE (as reflected in NOEM) and thus on the NO diurnal cycle. However, we must first address the question as
to why the NO magnitude is so much larger than the model. We first explore the possibility that the model atomic oxygen is
discrepant with observations.

Figure 5 shows the calculated atomic oxygen profiles for the equator from the driven and nudged models. Also shown are
two profiles from the Sounding of the Atmosphere with Broadband Emission Radiometry (SABER). The curve labed "old sab"

is the standard Version 2 product described by Mlynczak et al. [2013]. The curve labeled "new sab" represents a reprocessing
of that data using new kinetics as discussed by Mlynczak et al. [2018b]. Although the new SABER is lower than the old, both
profiles still exceed the driven and nudged models at all altitudes above 90 km and at 100 km, near the peak of the NO, the
difference is almost a factor of 3. This means that the driven and nudged models will necessarily underestimate the rate of
reaction (2). Since reaction (2) competes with reaction (1), underestimating reaction (2) means an overestimate of reaction (1)

and thus will lead to overestimating the production of NO. Therefore increasing the model $O$ should reduce the model $NO$. To
increase the model atomic oxygen, we ran the nudged model with Kzz (originally set to that used by Jones et al., 2017) divided
by a factor of 10. As discussed by Jones et al. [2017], Siskind et al., [2014] and earlier, by Forbes et al. [1993], reducing vertical
transport, in this case by reducing the diffusion, will reduce the recombination of atomic oxygen in the mesosphere and thus



lead to increased O in the lower thermosphere. Figure 5 shows that the atomic oxygen for the nudged-Kzz/10 case is about a factor of two greater than the other models and is within 30% of the new SABER values.

Figure 6 shows the diurnal variation of the calculated NO from the Kzz/10 model and also from the model where the reaction rate coefficient for (2) above was arbitrarily increased by a factor of 2. Both models show significantly less NO than the baseline

models shown in Figure 3. Doubling the rate of (2) was more effective in getting the NO magnitude down to the $4\text{-}5 \times 10^7$ range seen in SOFIE and NOEM. However, increasing atomic oxygen to agree better with SABER still did reduce the NO by about 50%. Importantly, both simulations still show the peak NO at sunrise with a lifting of the layer peak in late morning. Thus this diurnal variation is seen to be somewhat robust against changes in the NO chemistry. As we will discuss in the next section, the dawn peak is related to tidal oscillations.

**3.3  Diurnal variation**

The NO maxima in the TIME-GCM, whether at sunrise in the NAVGEM nudged model or at midnight in the driven model, are clearly associated with descent. This can be seen by looking at Figure 7 which shows the diurnal variation of three indicators of vertical motion in the lower thermosphere: the temperature, the $O/O_2$ ratio and the nitric oxide mixing ratio. All three of these quantities increase with altitude, thus a local increase at a single pressure level reflects vertical transport downwards

from higher altitudes. The assumption that NO can be treated as a passive tracer to study tidal variability was discussed by Oberheide et al. [2013]; it is particularly valid for the night time and dawn conditions discussed here where chemical damping is at a minimum. Figure 7 clearly shows that all three of these quantities show a peak near 6-7 AM local time in the nudged model. Further, by late morning these quantities decrease consistent with upwards motion. By contrast, in the driven model, they show a peak at midnight and minima at dawn. Comparing the vertical transport implied by Figure 7 with the calculated

NO densities in Figure 3, it is clear that the maximum in downward transport corresponds to the NO maxima shown in Figure 3. We can then further interpret the low altitude NO peak seen in SOFIE, relative to NOEM, as reflecting the diurnal variation of NO such that descent occurs in the early morning and this reverses in the late morning, when SNOE measured NO.

To understand the differences in the vertical transport in both models, we first note that a dominant mode of variability is a semi-diurnal oscillation. This is most evident in the driven model, but is apparent (and confirmed by spectral analysis, not

shown) in the nudged model. Thus we investigate the differences in the migrating semi-diurnal tide (SW2) in the two models. This is shown for temperature in Figure 8. It is clear that quite different solutions are reached between the two TIME-GCM models. In the driven case, the amplitude in the lower thermosphere (we use 108 km as a reference altitude) is quite large-about 25 K. This is well in excess of observations presented by Akmaev et al. (2008) and theoretical calculations presented earlier by Forbes and Vial (1991) whereas in the nudged case it is in good agreement with those references. Also apparent is a

distinct difference in the tilt of the lines of constant phase in the tide in the nudged case compared with the driven case. This is consistent with a change in the vertical wavelength such that it remains quite large in the driven case ($> 50$ km) but becomes smaller in the nudged case. As we show below, these differences are associated with a significant phase difference between the calculated SW2 in the driven and nudged models. We attribute these differences to the underlying zonal wind. This is shown in Figure 9 which presents the vertical profile of the zonal winds and the phase of the SW2 from the mesosphere to the lower




thermosphere. The differences in the zonal winds are consistent with that discussed by Jones et al. [2018] and is most likely attributed to a large eastward momentum source produced by the TIME-GCM's gravity wave drag parameterization. Figure 9b shows how the phase of the SW2 becomes increasingly different between the two models in the altitude region where the winds are different. At 108 km as illustrated by the short dashed line, this difference is about 6 hours, i.e. the SW2 in the nudged

model has become almost exactly out of phase with that in the driven model. This is completely consistent with the diurnal variation of vertical transport shown in Figure 7 where the descent in the driven model peaked near midnight and that for the nudged case, 6 hours later near sunrise.

Based upon theoretical work by Forbes (2000) and Forbes and Vincent (1989) on the effects of mean winds on atmospheric waves, we understand that both the phase and amplitude of the SW2 tide will be sensitive to the background zonal wind field

through which it propagates. Specifically, as discussed by Forbes (2000) a wave propagating in the opposite direction of the mean wind, as is the case here, will see an increased vertical wavelength and decreased damping. Thus the greater eastward zonal winds seen in the driven model will correspond to a longer wavelength in the SW2 tide and to a greater amplitude. Since the winds in the driven case are much greater than observed, it stands to reason that the amplitude of the tide is much greater than observed.

## 4   Discussion

Our model/data comparison has implications for our understanding of both the dynamics and the chemistry of the MLT. First, our work illustrates the need for an accurate simulation of the mesospheric zonal winds in calculating the diurnal variation of NO. It is probably not surprising that the simulation of the mesospheric zonal winds affects the propagation of tides up from the middle atmosphere. Jones et al. (2018) cover aspects of this topic using the TIME-GCM with different nudging scenarios

and the theoretical studies we cited above have presented analytic explainations for several decades. However, in illustrating how variations in the migrating semi-diurnal tide can affect the variation of nitric oxide, we have presented a new mechanism for whole atmosphere coupling from the middle atmosphere to the thermosphere.

We should note that although we have discussed the semi-diurnal tide, the transport effect is not apparent in the model at sunset. We interpret this as due to the importance of daytime chemistry. Thus all our models show a distinct difference in the

sunrise and sunset abundances. This is important because satellites such as ODIN are in sun-synchronous orbits and acquire both sunrise and sunset data. The recent ODIN-based model of Kiviranta et al [2018] did not distinguish between sunrise and sunset data; our results suggest it would be useful to do so.

More problematical than the diurnal variability might be the question of the absolute abundance of the calculated nitric oxide which is significantly overestimated by the model. Interestingly, Hendrickx et al. [2018] encountered the same problem with the

Whole Atmosphere Community Climate Model (WACCM) compared with SOFIE in the auroral zone. Hendrickx et al. [2018] also mentioned the importance of capturing the correct compositional abundances of the background atmosphere, specifically the atomic oxygen. Here we show that tuning the model to better match SABER, while certainly necessary, appears insufficient. Even with $O$ being in reasonable agreement with SABER, the model still significantly overestimated the $NO$. Conventionally,





in this situation, the other unknown that NO modelers focus on is the solar flux (Siskind et al., 1995; 1990), specifically the soft X-rays that ionize and dissociate $N_2$ in the lower thermosphere. However, in this case, the presumed remedy to reduce the model NO would be to reduce the soft X-ray flux and this would likely make the calculation of the E-region electron densities worse. The reason is illustrated in Figure 10 which shows three TIME-GCM model simulations compared with IRI, all for

noon at the equator. Note how below 125 km, where soft X-ray ionization becomes important, the models all underestimate IRI. This is a robust comparison because a new empirical model, the Faraday-IRI, 2018 model of Friedrich et al. (2018) shows E-region electron densities using a different dataset than used by IRI and gets similar answers, i.e. for high sun conditions at 100 km, the electron density equals or exceeds $10^5$ cm$^{-3}$. This model underestimate of the E region electron density has been recognized before (Maute, 2017 and references therein) and is the reason that those authors increased the soft X-ray flux.

Pavlov and Pavlova (2015) had the same problem and made the same change to the reference soft X ray spectrum, i.e. they increased it. Thus it appears that the requirements of the nitric oxide simulation and the E-region electron density simulation are in conflict.

To resolve the model overestimate of nitric oxide without worsening the model underestimate of E-region [e-], we can suggest two possibilities. The first is that aspects of the nitric oxide kinetics should be reevaluated. We have shown that

increasing the $N(^2D) + O$ rate can bring the model NO into agreement with SOFIE and NOEM. As we discussed above concerning equation (2), the evidence for this particular rate coefficient being underestimated by a factor of 2 is mixed at best; our adoption of this faster rate could be best considered as a proxy for other, as yet unidentified, changes to the odd nitrogen kinetic scheme. A second possibility is perhaps more speculative but is intriguing in that it might solve two problems at once. This would be a scenario where by the E-region production of $O_2^+$ is increased. The $O_2^+$ could then serve as a partial sink for

nitric oxide via

$$O_2^+ + NO \rightarrow NO^+ + O_2 \tag{4}$$

The sink is described as only "partial" because the recombination of $NO^+$ favors $N(^2D)$ so that much of the NO will be reformed (Yonker, 2013). However, it could go in the right direction. Conventionally, much of the $O_2^+$ production in the E-region is from the strong solar Lyman $\beta$ line. As with the rest of our solar spectrum, we use Solomon and Qian (2005) which

for the band which includes Lyman $\beta$ (987.7 - 102.7 nm) gives a near solar minimum flux of about $4.5 \times 10^9$ cm$^{-2}$s$^{-1}$. This is unlikely to be an underestimate when compared with other datasets such as that discussed by Warren et al. [2005, 1998] so we are constrained from making a drastic change (such as a doubling) to the flux at these wavelengths. An alternative idea was proposed by Meier et al. (2007) who suggested that the use of averaged cross sections could have the effect of underestimating the penetration of some of the solar EUV spectrum down to 110 km. They show that the $O^+$ ionization in the E region could

be significantly increased with higher resolution. Interestingly, the wavelengths they discuss are those which ionize $O_2$, but not $N_2$. Thus although Meier et al. (2007) did not show $O_2$ ionization, it seems plausible that their high resolution cross sections would lead to more $O_2$ ionization in the E region. This should be given some consideration for future work.



## 5 Conclusions

Taken together the indirect comparison of SOFIE with NOEM shows that we can reasonably define a baseline minimum value of NO, approximately equal to $4 \times 10^7$ cm$^{-3}$, which represents solar minimum conditions. Further we have provided some insight into the possible reason for the approximate 8 km altitude offset between the peak NO seen by SOFIE and NOEM. This

difference appears to be dynamically driven, specifically due to the 6 hour phase change of the migrating semi-diurnal tide as it propagates up from the stratosphere to the lower thermosphere. While the TIME-GCM as nudged by NAVGEM does not give an exact replication of the sunrise/11 AM NO ratio indicated by the SOFIE-NOEM comparison, it does support the existance of a low altitude NO peak at dawn. Our results further suggest that a more accurate simulation of the zonal wind would likely give a better simulation of the sunrise/11 AM NO ratio.

Our calculation of the absolute abundance of nitric oxide significantly exceeds the SOFIE/NOEM data. What is new to our approach here is that we simultaneously compare our calculation to the SABER atomic oxygen data as well as to empirical models of the E-region ionosphere. This kind of comparison significantly constrains the number of free parameters; in the case where we tune the model to improve agreement with SABER, we also improve the agreement with SOFIE/NOEM. However, lowering the soft X-ray flux to further reduce the NO would likely worsen the model underestimate of [e-]. We thus suggested

an alternative, admittedly more speculative, remedy involving use of higher resolution cross sections to increase E-region ionization. Regardless of the specific remedy to the discrepancy, our study points to the value of a multi-constituent approach (i.e. $O$, $NO$, and [e-]) towards validating models such as the TIME-GCM and demonstrates the utility of nitric oxide as a useful diagnostic of chemical and dynamical processes at the base of the thermosphere.

*Data availability.*   The SOFIE NO data can be obtained by ftp from gats-inc.com/sofie. NOEM is distributed as part of the NCAR/GLOW

model( Stan Solomon, PI) and can be downloaded from https://github.com/NCAR/GLOW. Daily NCAR TIME-GCM outputs in netCDF format from this study are archived on the DoD HPCMP long-term storage system. The NAVGEM-HA inputs used to constrain the TIME-GCM simulations presented here have been submitted for approval from the US Navy for public release for availability after this paper has completed being reviewed.

*Author contributions.*   DES did the model/data comparisons and wrote the main body of text. MJ provided the model results, did the SW2

analysis, and helped edit the text. DPD assisted in configuring the TIMEGCM simulations and consulted on the TIMEGCM nudging scheme. JPM consulted on the NAVGEM model results, their range of validity and comparison with radar winds. MEH consulted on the SOFIE data and its range of validity and helped edit the text. DRM assisted with the NOEM code and consulted on the model/data comparison. MGM consulted on the range of validity of the new SABER $O$ data and helped edit the text. SMB consulted on the comparison of NOEM and SOFIE and help edit the text. AM consulted on the use of the TIMEGCM to calculated electron density. NJM consulted on the use of the

Ascension Island wind data.



*Competing interests.* The authors declare that they have no conflict of interest.

*Acknowledgements.* We acknowledge support from the NASA AIM Small Explorer program (through Interagency Purchase Request S50029G to NRL), the NASA/TIMED SABER project (through Interagency Purchase Request NNG17PX04I to NRL) and the Office of Naval Research BSION program, award number N0001417WX00579. Additionally, DPD acknowledges support from the NASA Heliophysics Supporting Research (HSR) program through interagency agreement NNH17AE69I to NRL. This work was performed while M. Jones Jr. held an NRL Karles Fellowship. JPM acknowledges support from NASA grant NNH13AV95I. AM is supported by NASA grants X13AF77G and NNX16AG64G. Computational resources for this work were provided by the U.S. Department of Defense (Dod) High Performance Computing Modernization Program (HPCMP). We thank Stan Solomon of the High Altitude Observatory for helpful discussion concerning nitric oxide chemistry. NCAR is supported by the National Science Foundation.



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





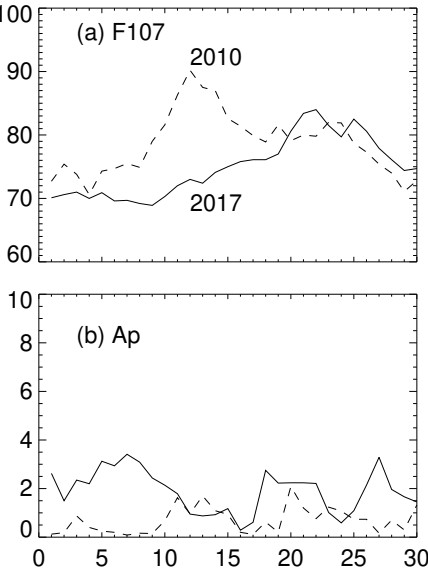

**Figure 1.** (a) Daily variation of the solar F107 index for January 2017 (solid) and 2010 (dashed). (b) Daily Ap for the same period.

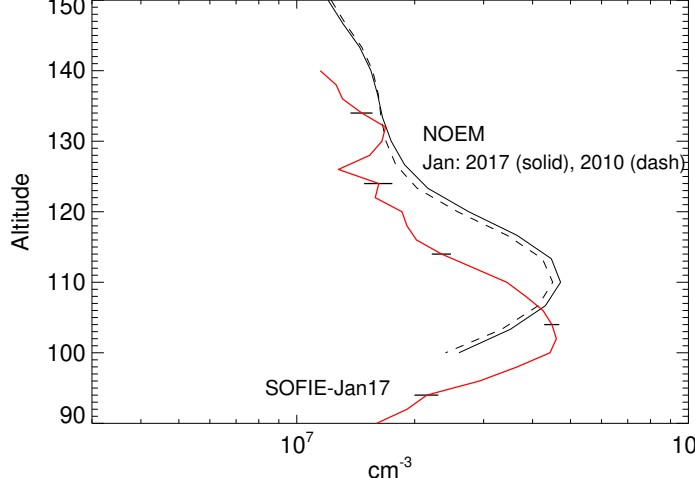

**Figure 2.** Averaged January 2017 SOFIE NO profile (red) compared with the Nitric Oxide Empirical Model (NOEM) for January 2010 and January 2017.

[t]





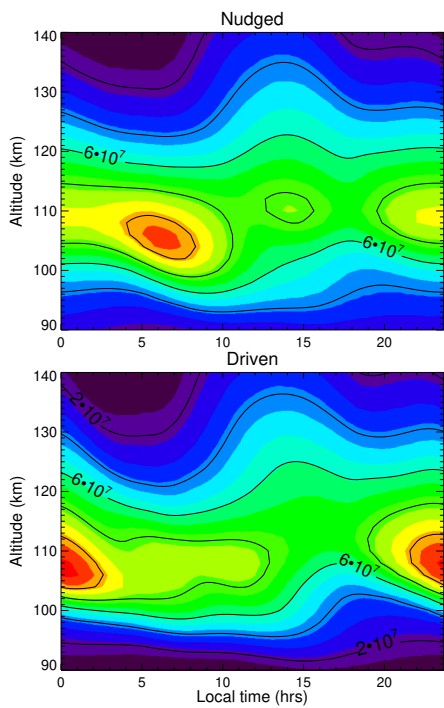

**Figure 3.** Diurnal variation of calculated nitric oxide at $4°$ S. Top: the model nudged by NAVGEM-HA and bottom: the model driven at the bottom boundary by the GSWM model and the MERRA analysis. The contour interval is $2 \times 10^7 \mathrm{cm}^{-3}$.





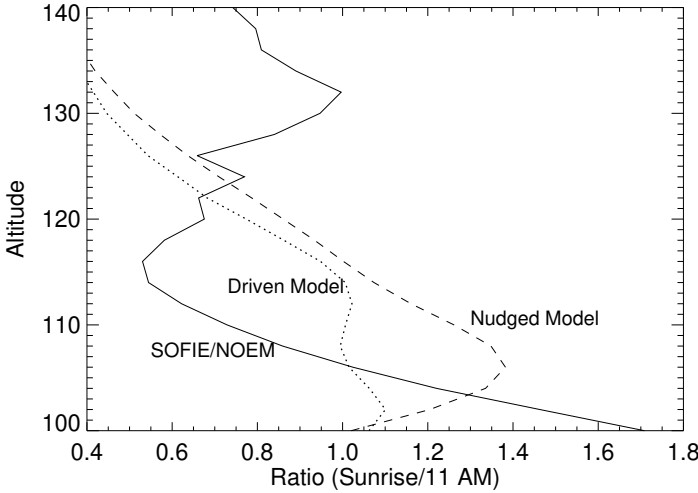

**Figure 4.** Ratio of sunrise to late morning nitric oxide. For the two models, this was done by taking the ratio of the nitric oxide averaged from 0500-0600 local time (to approximate the local time of the SOFIE observations) to that at 1100 local time (roughly the local time of the SNOE observations). Also shown as the solid line is the ratio of the SOFIE data to the NOEM 2017 model shown in Figure 2.

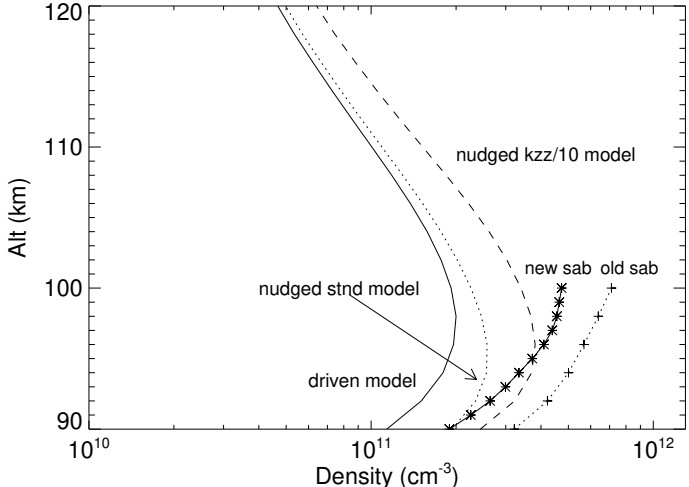

**Figure 5.** Monthly averaged atomic oxygen profiles from three simulations with the TIME-GCM and two versions of the SABER database. The curve labeled "old sab" is the Version 2 data of Mlynczak et al. [2013]; the curve labeled "new sab" is the reprocessed data described by Mlynczak et al., [2018].





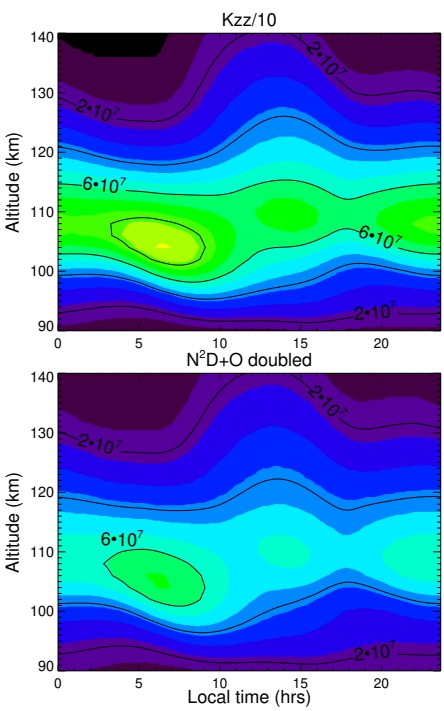

**Figure 6.** Diurnal variation of calculated nitric oxide at $4°S$, both with the nudged model, but with the additional indicated changes to the TIME-GCM.

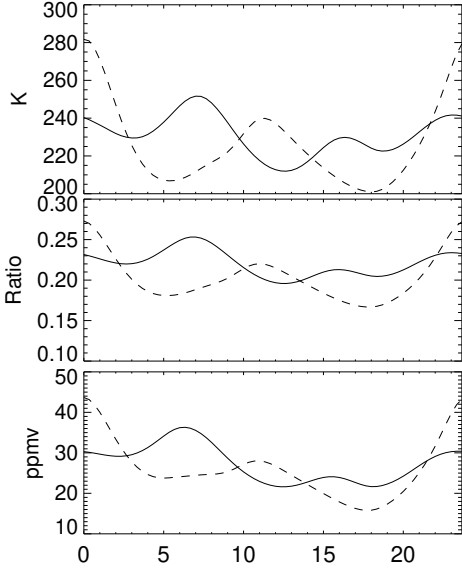

**Figure 7.** Time variation of (top) temperature (middle) $O/O_2$ ratio, and (bottom) nitric oxide mixing ratio at $p=1.1 \times 10^{-4}$ hpa (about 108 km). The solid lines are for the nudged model, the dashed lines are with the driven model.





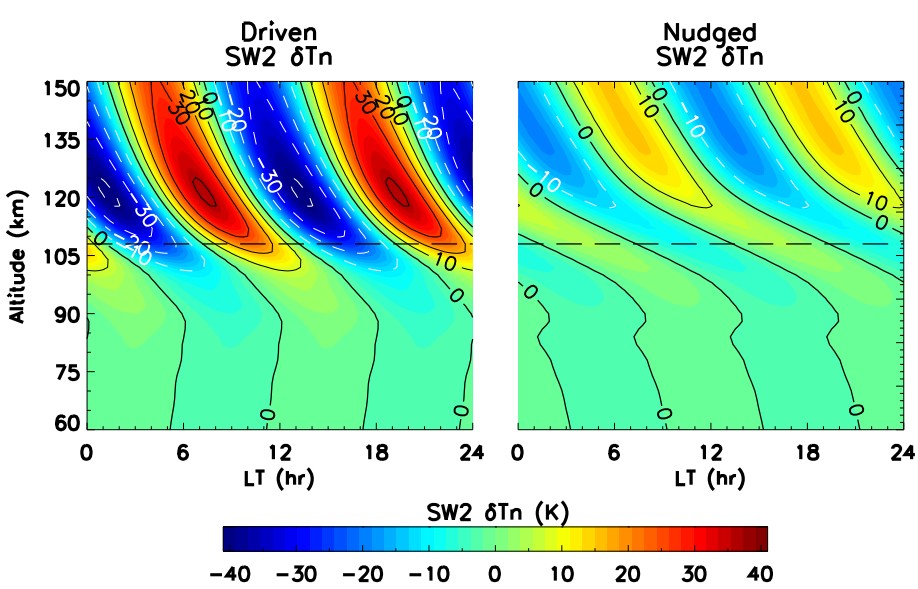

**Figure 8.** Variation of SW2 temperature tide vs altitude for (left) the GSWM driven model and (right) the NAVGEM nudged model



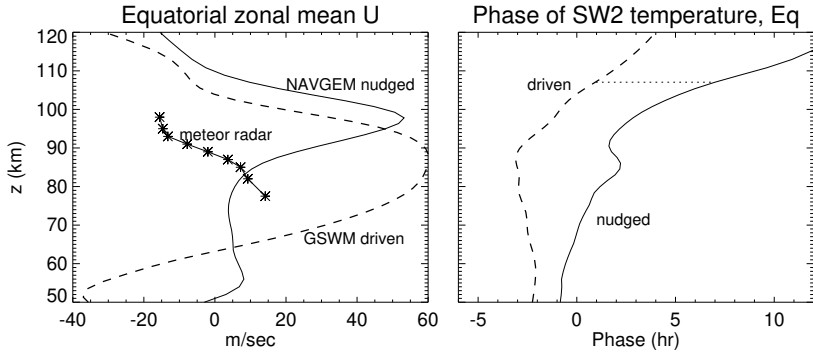

**Figure 9.** (a) Vertical profiles of TIME-GCM zonal winds for the GSWM driven model (dashed line) and the NAVGEM nudged model (solid). Also shown for comparison are monthly averaged meteor radar winds for January 2010 taken from Ascension Island (8S, 14.4W) (b) Associated phase of the SW2 tide for the two TIME-GCM simulations. The thin dotted line is simply a reference fiducial to show the phase difference between the two simulations at 108 km

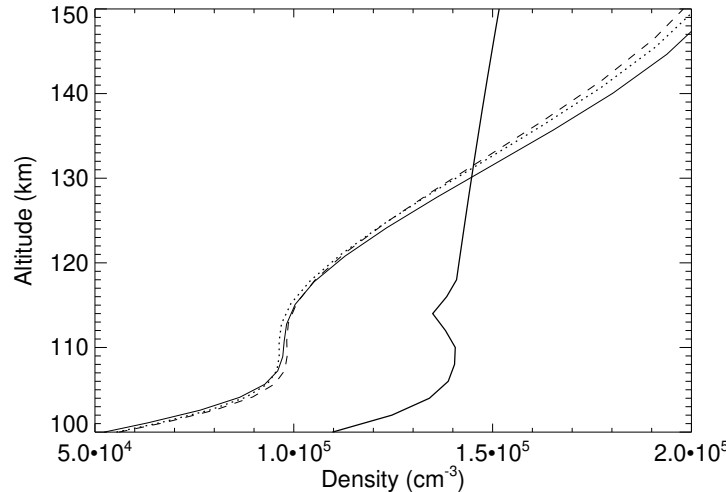

**Figure 10.** Altitude variation of electron density from three TIME-GCM simulations as compared with IRI, all for noon at the equatori. The thick solid line is IRI, the thin solid line is the standard nudged model, the short dotted line is the nudged with Kzz/10 model, and the dashed line is the nudged model with the reaction rate coefficient for $N(^2D) + O$ doubled.