# Peer review of "On the relative roles of dynamics and chemistry governing the abundance and diurnal variation of low latitude thermospheric nitric oxide"

_Annales Geophysicae, 2018_

## Referee Comment (RC1) · K. Hendrickx (Referee) · 25 Oct 2018

Review of 'On the relative roles of dynamics and chemistry governing the abundance and diurnal variation of low latitude thermospheric nitric oxide' by Siskind et al., submitted to Annales Geophysicae.

General comments:
The work by Siskind et al. uses several runs of the TIME-GCM model (nudged to NAVGEM or driven with GSWM/ECMWF) to provide possible explanations for the observed differences of equatorial nitric oxide between AIM/SOFIE and SNOE

(via NOEM) satellite observations. The semi-diurnal tide is found to be responsible for the altitudinal difference between the maximum layers of NO in both satellite derived datasets. Absolute abundance differences between the model and satellites are improved using atomic oxygen profiles similar to SABER observations and by increasing the rate coefficient of a net-reducing-NO reaction.

In general, the manuscript is well structured and reads fluently. Some minor suggestions are given to provide more information and improve figure clarity. The References section needs some improvement. The results and conclusions are supported by the data and the topic is within the scope of Annales Geophysicae. This work is recommended for publication after the following suggestions are implemented.

Specific comments:

p3, first paragraph of Section 2:
- in the description of the SOFIE observations, please specify which latitudes are observed in December 2016-January 2017 and at which local times. Before this paragraph it is stated that SOFIE observes both sunrise and sunset, while later in this section the 5-6 AM hours are given. This paragraph might be ideal to provide some more information.
- when first reading the sentence on line 11, I understood 'both months' as the months December and January, even though they correspond to January 2010 and 2017. Please make this more clearer.
- on line 12, it is stated that Figure 1 shows that solar activity was slightly higher than absolute solar minimum. However, this statement cannot be made from the figure. Please also clarify what is considered as absolute solar minimum: solar cycle 24 began in December 2008.
- please provide a one-sentence introduction on the solar F10.7 radio flux and the

planetary Ap geomagnetic index on line 12.
- on line 13, it is stated that the averaged Ap is 9 for January 2017 and 3 for 2010. This does not correspond with what is seen in Figure 1 (b), where I would say an average Ap is seen of 1 for 2010 and 2 for 2017.

on the TIME-GCM model:
- overall, I think the information given about the model is very limited and scattered over different sections, making it difficult for the reader who is not familiar with TIME-GCM to obtain an overview of the model's capabilities and limitations. Could a paragraph, which probably fits best in Section 2, be included with some more basic information about the model as well as references? What is for example the upper altitude limit of TIME-GCM and vertical spacing?

p4,l21-24: I find this to be a weird phrasing: what is the importance of the Herron (1999) paper? And if Herron recommends a rate coefficient determined at room temperature, then at which temperature was the Fell rate coefficient determined? Please rewrite.

p5,l10-17: in this paragraph the dynamical and chemical lifetimes of NO could be discussed. The Marsh et al. paper (A tidal explanation for the sunrise/sunset anomaly in HALOE low-latitude nitric oxide observations, GRL, 2000), which should be cited in this study, shows that the dynamical lifetime will control NO variations below about 110 km, while chemistry will dominate above. This is similar to the 115 km altitude, where both models break from the dynamical to the chemical controlled regime.

p6, paragraph from l4-9:
- was the rate coefficient increased or multiplied by a factor of 2? And is 'arbitrarily' the

right word to use here if it was suggested by Herron et al. (1999)?
- one can only compare Figure 6 to the nudged model in Figure 3, so rewrite perhaps to 'Both simulations show significantly lower concentrations than the NO baseline in the nudged model shown in Figure 3.'
- write 'down to the 4-5 X $10^7 cm^{-3}$ range near the maximum NO density, as seen in SOFIE and NOEM.'

p6,l26: please elaborate a bit more on what is shown in Figure 8, how it is calculated (T variations from which background?) and what is meant by 'This'.

p6,l34: the vertical profile of zonal winds, at which latitudes?

p7, 32: please provide a small discussion on how realistic the used approach is to obtain better agreement of the [O] profile between models and SABER.

p8,l4: please define the abbreviation IRI at first occurrence
p9,21: similar for DoD and NAVGEM-HA

Technical comments:

Several articles that are referenced in the text are not included in the references section:
p2,l3: Kockarts (1980), Mlynczak et al. (2003), Knipp et al. (2017)
p2,l8: Smith-Johnson et al. (2017)
p2,l9: Randall et al. (2015)
p2,l17: Barth et al. (1998)

p4,l4: McCormack et al. (2017)

On the References section:
p11,l5: write 'Venkataramani' instead of Ventkataramani
p11,l6: write 'sudden' instead of suddent
p11,l12: missing DOI
p11,l26: write a comma before the start of the article's title, missing DOI
p11,l35: write 'Païvärinta', provide the full title, missing DOI
p12,l3: write 'Gómez-Ramírez', missing DOI
p12,l6: write 'Observation' instead of Observations
p12,l8: missing DOI
p12,l16: write 'Pérot'
p12,l18: write the full title
p12,l24: missing DOI
p12,31: write the full title
p13,l1: missing DOI
p13,l29: missing DOI
p13,l31: write Å instead of A
p13,l32: write observations instead of 'observatiosn'

On the Figures:
Figure 1: no y-axis labels and units are provided
Figure 2: no y-axis unit, no x-axis label, what are the black lines on the SOFIE profile?
Figure 3: NAVGEM-HA has not been defined, providing a colour bar could be useful for the reader to infer NO concentrations throughout the altitude region and aid in the comparison to the SOFIE and NOEM profiles (especially since no nitric oxide profiles are shown from the models), the black contour line labels are not very readable (perhaps white is more clear)

Figure 4: no y-axis unit
Figure 5: write 'Altitude' on the y-axis for consistency, 'stnd' has not been clarified in the caption.
Figure 6: similar remark for the contour lines as in Fig. 3, providing a colour bar or perhaps showing the NO variations from the nudged model NO baseline would give more information for the reader.
Figure 7: no y-axis labels (T [K], Ratio O/O2, NO [ppmv]), write 'dashed lines are for the driven model'
Figure 9: write 'Altitude' as y-axis label for consistency, write a full stop in the last sentence of the caption
Figure 10: write 'equator' instead of equatori

Typos and suggestions:
p2,l10: mesosphere instead of mesospheric
p2,l27: the Marsh et al. paper was published in 2004
p4,l19: calculated instead of calcuated
p4,l26: I believe it should be 'peak NO values in the lower thermosphere'
p5,l4: write 'slope in the nudged model', 'between 105 and 115 km'
p5,l5: write 'Just as in the observational data, the nudged model ratio decreases'
p5,l24 and l25: write labeled instead of labed
p7,l1: write 'with those discussed . . . and are most likely'
p7,l16: write 'of the equatorial MLT'
p7,l20: explanations instead of explainations
p9,l2-3: write ' a baseline minimum value of the NO peak', 'represents solar minimum conditions near the equator.'
p9,l25 and l29: write TIME-GCM
p9,l29: write 'and helped edit'

---

## Referee Comment (RC2) · Anonymous Referee #2 · 5 Nov 2018

The manuscript objective is to study the relative roles of dynamics and chemistry on thermospheric mean nitric oxide (NO) and its diurnal variation during solar minimum at low latitudes with emphasis on the migrating semidiurnal tide (SW2) and existing conflicts of NO and electron density modeling in the E-region. The approach is to use a combination of January 2017 SABER/TIMED atomic oxygen and SOFIE/AIM NO data along with 2010 TIME-GCM model simulations (free running and nudged to NAVGEM< 95 km) and comparisons with SNOE data (through the NOEM empirical model). The main findings are as follows: (i) mesospheric zonal mean zonal winds are important to understand the diurnal NO variation because they impact the SW2 tidal magnitude and phase and thus the related downward transport of thermospheric

[Figure]

NO, (ii) tuning the model atomic oxygen towards SABER still results in a too high NO and the standard approach to decrease the soft X-ray flux would drive TIME-GCM electron densities even further away from IRI, pointing to an existing conflict of the NO and electron density model requirements. A suggested remedy is an increased O2+ production (which serves as a a partial sink for NO) as a result of higher wavelength resolution of the EUV spectrum and its penetration down to 110 km (O2 ionization but not N2).

NO is important for the infrared cooling of the thermosphere, as a source of NO+ ions, and as a measure of energetic particle precipitation into the atmosphere. The relative roles of dynamics and chemistry using observations have been understudied before due to limitations with the local times of the available NO data. Part of the presented work is a comparison of the SNOE-based empirical NOEM model and SOFIE which confirms the existence of an altitude offset between the two data sets explained by the semidiurnal tide. Although speculative, the proposal that higher resolution EUV spectra might be needed to remedy NO issues in the model is interesting and reasonably well motived. The manuscript is therefore relevant for solar-terrestrial physics and to the ANGEO readership. I find it well well-written and fully recommend publication once a few minor comments have been addressed. Comments 1, 3 and 6 are the most relevant ones.

1. TIME-GCM is apparently from 2010 runs but used to interpret Jan 2017 data. An argument is made on page 3 that the NO profiles shouldn't differ much from January 2010 to January 2017. It is, however, not clear from the manuscript why this is the case. The manuscript highlights the importance of the mesospheric zonal mean zonal winds for the SW2. Was the SW2 in the data the same during both years? This seems to be quite important for the conclusions and can easily be checked using SABER temperatures, for example.

2. While Figure 3 show the altitude/local time variation of the modeled NO as a contour plot, I would find Figure 2-style line plots for the model helpful, as an additional Figure.

It would more clearly show the data/model differences.

3. What is the error bar of the SOFIE NO? I believe this information is needed to help the reader with his/her assessment of model/data differences. This comment is not only in the light of the Figure 4 discussion above 115 km, but as a general issue for any model/data comparisons.

4. Page 6, line 27-29. What are the reference values?

5. Figure 9. What is the purpose of the Ascension Island MR winds? They are not discussed in the text and even the wind direction is different. There's only a vague reference on page 7, line 13.

6. Page 7, line 24. The discussion of the sunset/sunrise differences is too vague and the argument regarding the importance of daytime chemistry is too hand-waiving. What chemistry and how/why is this consistent with the ss/sr differences? Also, why is it not related to or impacted by the migrating diurnal tide (DW1)? The latter is in a different phase at sunset. Could this explain the difference, e.g., SW2 and DW1 work together during sunrise but against each other during sunset?

7. Figure 10. Typo in the caption and the line thickness of IRI seems to be the same as for TIME-GCM.

8. Overall, all Figures should be checked for axes, labels, etc.

---

## Referee Comment (RC3) · Anonymous Referee #3 · 21 Nov 2018

This work employs the TIME-GCM model in nudged and driven configurations together with various observational and observation-derived datasets to investigate equatorial lower thermospheric nitric oxide in terms of absolute magnitude and local time dependence. The novelty of the approach resides in the simultaneous comparison of observed and modeled key parameters ruling nitric oxide abundances, such as atomic oxygen and ionospheric parameters, allowing to test model implementations of chemical and dynamical processes at the base of the thermosphere.

The paper is well written and the topic is relevant for Annales Geophysicae. It certainly merits publications, subject to a few minor comments and suggestions listed below.

[Figure]

p 2 l16-19 SCIAMACHY on ENVISAT also measured thermosphere NO (Bender et al. 2013, www.atmos-meas-tech.net/6/2521/2013/). In this context it should also be noted that there are several recent empirical models of lower thermospheric NO from satellite observations with different local time sampling that might have been useful in addition to the employed NOEM model, e.g., the SANOMA model based on SMR data (Kiviranta et al, 2018, https://doi.org/10.5194/acp-18-13393-2018) or the SCIAMACHY-based model of Bender et al., ACPD, https://doi.org/10.5194/acp-2018-872).

p 3 l14 Figure 1 seems to display Kp rather than Ap. For instance, on 20 Jan 2010, Ap was 12 and Kp 2, the latter much better in line with the corresponding value of the dashed line in Fig. 1. The average Ap values quoted at l14 seem, however, to be correct.

Figure 9: The meteor radar data shown in panel (a) is not discussed in the body text.

Technical:

p 2 l10 mesospheric -> mesosphere

p 2 l27 1100 -> 11:00

p 8 l19 remove "by" before "the E-region"

---

## Author Comment (AC1) · 21 Dec 2018

We thank the reviewer for his comments and careful read of the manuscript and apologize for the myriad of typos in the reference list. We have tended to most all of the suggested editorial changes, including the addition of a small section describing the TIME-GCM. In particular, we thank the reviewer for catching our error in plotting the Ap index in Figure 1b (we had Kp, not Ap). We do not label the axes since they are unit-less, but we added a line in the text on page3 (line 12) stating where we obtained the values. We clarified the solar/geophysical variability by giving reference to a US government web site which displays the variation of F107 and Ap over the last 11

years (https://www.swpc.noaa.gov/products/solar-cycle-progression). We also clarified the differences in the literature concerning N(2D) quenching by O, added the Marsh and Russell GRL (2000) citation, added a sentence to support the approach we used to improve the O-atom agreement with the data, explained how Figure 8 was produced and added an explanation of the zonal wind comparison in Figure 9. We also cleaned up the reference list and added the requested axis labels and color bar to the figures.

---

## Author Comment (AC3) · 21 Dec 2018

We thank the reviewer for his/her comments. Our specific response is as follows:

Regarding the two references, we already reference and discuss Kiviranta in Section 4. The Bender model is, as of this writing, still under review. Further, SCIAMACHY measured at a local time near SNOE so it should be similar to NOEM. We thus have declined to add this reference.

Yes, the reviewer is correct about Figure 1- we apologize and thank the reviewer for catching it; it was our error and is now corrected.

[Figure]

Figure 9: we have added discussion about the meteor wind data.

The technical edits have been implemented. We note that the word "by" was supposed to be part of one word "whereby".

---

## Author Response (AR1)

Reviewer 1:

We thank the reviewer for his comments and careful read of the manuscript and apologize for the myriad of typos in the reference list. We have tended to most all of the suggested editorial changes, including the addition of a small section describing the TIME-GCM (page 4). In particular, we thank the reviewer for catching our error in plotting the Ap index in Figure 1b. We do not label the axes since they are unit-less, but we added a line in the text on page3 (line 12) stating where we obtained the values. We clarified local times and latitude of the SOFIE data (page 3). We clarified the solar/geophysical variability by giving reference to a US government web site which displays the variation of F107 and Ap over the last 11 years (https://www.swpc.noaa.gov/products/solar-cycle-progression). We also clarified the differences in the literature concerning N(2D) quenching by O (page 5), added the Marsh and Russell GRL (2000) citation (page 7), added a sentence to support the approach we used to improve the O-atom agreement with the data (pagd 6), expanded the discussion of how Figure 8 was produced (page 7) and added an explanation of the zonal wind comparison in Figure 9 (page 7). We also cleaned up the reference list and added the requested axis labels and color bar to the figures. Note that we clarified DoD and NAVGEM, but that the IRI abbreviation was already defined at the end of section 1.

Reviewer 2:

We thank the reviewer for his/her comments. Our specific response is as follows:

1. That the overall magnitude of the NO density at the equatorial is predominantly due to solar forcing is well established from SNOE data as noted in the text on page 3 (line 19) is consistent with the near identical results from NOEM for the two January's. Nonetheless, the idea that the SW2 could change given the differing wintertime dynamics between 2010 and 2017 is important to acknowledge and we have added some discussion in Section 4 to cover that, as well as a citation to Pedatella and Liu (2013). There was an SSW in 2010 that did not occur in 2017; however, the 2010 event was at the end of our averaging period, so overall the two months' dynamics were unlikely to be that different. It is not something that can be easily checked in observations, particularly since SABER local times vary and it is typically only used to give 60 day averages. We also checked and mention, but do not show, for phase changes in other tidal modes (DW1 and TW3) and did not find any.
2. Regarding line plots- given the over factor of two discrepancy in absolute abundance between TIME-GCM and data, we do not feel that comparing line curves would be helpful and thus declined this suggestion.
3. The figure already gave geophysical error bars, we had just neglected to explain that in the caption; this is now added. The SOFIE curve in Figure 2 represents an average of over 400 profiles (15 x 30 days); since Table 1 of Gomez-Ramirez shows that signal profile uncertainties in the 90-120 km altitude range are no more than 25%, these random retrieval errors will be reduced to insignificant values in a monthly average. Discussion of this has been added to the text in Section 2.
4. We have added discussion about how Figure 8 was constructed.
5. We have added discussion of the purpose of the radar winds in Figure 9.

6. Regarding the daytime chemistry, we have added more discussion (Section 4) about the relative roles of dynamics and chemistry to hopefully clarify things.
7. And we have cleaned up the figures.

Reviewer 3:

We thank the reviewer for his/her comments. Our specific response is as follows:

Regarding the two references, we already reference and discuss Kiviranta in Section 4. The Bender model is, as of this writing, still under review. Further, SCIAMACHY measured at a local time near SNOE so it should be similar to NOEM. We thus have declined to add this reference.

Yes, the reviewer is correct about Figure 1- we apologize; it was our error and is now corrected.

Figure 9: we have added discussion about the meteor wind data.

The technical edits have been implemented. We note that the word "by" was supposed to be part of one word "whereby".

[revised manuscript text omitted]